# GRADIENT-BASED TRANSFER LEARNING

## ABSTRACT

We formulate transfer learning as a meta-learning problem by extending upon the current meta-learning paradigm in that support and query data are drawn from different, but related distributions of tasks. Inspired by the success of Gradient-Based Meta-Learning, we propose to expand it to the transfer learning setting by constructing a general encoder-decoder architecture that learns a map between functionals of different domains. This is achieved by leveraging on the idea that the task-adapted parameters of a meta-learner can serve as an informative representation of the task itself. We demonstrate the proposed method on regression, prediction of dynamical systems and meta-imitation learning problems.

## 1 INTRODUCTION

The ability to quickly adapt to unseen conditions is a necessary skill for any intelligent system. It provides the means to generalize outside of the training conditions as well as the capacity to extract unobservable features affecting the learner (Lake et al. (2017)). Adaptation to a new task involves two steps. The first is inferring the characterizing information of the task at hand. The second is regressing the function representing the task. The importance of this ability is reflected in the considerable volume of work conducted on the matter in the past years e.g. Hospedales et al. (2021); Ben-David et al. (2006); Ljung (2010). The field of meta-learning provides the means to unify these two steps and learn them simultaneously and fully data-driven (Huisman et al. (2021)). The learning process comprises multiple datasets representing different conditions, or tasks, the learner is concurrently exposed to. Adaptation is performed by extracting the relevant information about each task from a small set of data sampled from the task.

In this paper we consider the case of transferring knowledge using a small set of data from a task to another, different, task. In this regard, we build upon the framework of few-shot learning (Wang et al. (2020)). This can be summarized as estimating an optimal learner for any task with the fewest data samples possible. Recent work has explored the case where the data used for the adaptation and the downstream-task's data are subject to a distributional shift in their domain, referred to as *support-query-shift* (Bennequin et al. (2021)). Here, we assume the more general formulation of meta-transfer where the shift can take place on both the domain and co-domain of the underlying function generating the data. This brings us beyond the problem of domain-shift and into the more general notion of learning to transfer between support task and query task.

The need for transfer emerges in a multitude of situations. Sequential decision-making problems are one of them. Real-world dynamical systems, for example, are often only partially observable. They require an initial exploration phase to gather the necessary information before estimating a suitable policy. In this case, we would need a way to transfer the knowledge acquired from the dynamics of the system to the estimation of the target policy. That is, transfer between a dynamics prediction model to the estimation of a policy in a control problem. Moreover, transfer learning can be used in situations where we have access to labeled data of a simple problem but would like to solve a more complex, but related, problem. For example, transfer from a single inverted pendulum to a double pendulum with the same dynamics e.g. same length of the poles, same gravity and friction coefficients.

To this end, we present an approach to transfer learning through adaptation. Inspired by Gradient-Based Meta-Learning (GBML) we propose a method for meta-transfer learning in a general encoder-decoder model. This can be used independently of the shift between the support task and the query task and is agnostic to architectural changes between the meta-learner and the base-learner (see Fig-

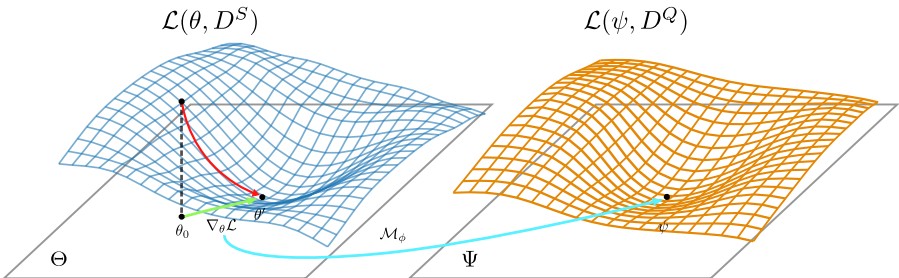

Figure 1: Visual depiction of the proposed model. The representation of the task is the gradient, $\nabla_\theta \mathcal{L}$, of a meta-learner model i.e. green arrow on the blue loss surface on the left. This representation is then mapped through $\mathcal{M}_\phi$ (light blue arrow) to the parameters, $\psi$, of the main network which is optimal for the given task (minimizes the orange loss function on the right)

ure 1). The main idea of this work is that the parameters of a learner that is optimal for a given task contain all the relevant information of such task (Tegnér et al. (2022)). The proposed model learns a map from the gradients used to adapt the parameters of a meta-learner to the parameters of a base learner. This, in fact, is a map between functions that has proven to be effective in different contexts e.g. Xu et al. (2020); Dupont et al. (2022). We argue that representing the task's parameters as the gradients of the meta-learner is more robust to noise and bias in the data. We empirically support this claim with a number of experiments on synthetic regression, dynamical system prediction and meta-imitation learning.

Our contributions are as follows:

- We extend the formulation of support-query shift to the problem of transfer learning.
- We describe a meta-transfer learning method that builds upon previous gradient-based methodologies.
- We provide an empirical evaluation of the advantages of gradient-based task representations on a variety of problems.

## 2 RELATED WORK

In this section we review the relevant literature regarding adaptation methods. The idea of adapting the learning system has been widely studied in the past years (Naik & Mammone (1992); Bengio et al. (1990); Hochreiter et al. (2001)). Adaptation is performed using data-points that uniquely characterize the task. Different approaches can be included in this definition depending on the framework they abide to and the assumptions they make about the adaptation process.

**Transfer Learning.** It refers to the problem of learning algorithms to extract knowledge from one task to solve a second task (Weiss et al. (2016); Zhuang et al. (2020); Pan & Yang (2009)). These methods cannot be considered to perform general adaptation strategies. However, they require the identification of useful information for a task from a different distribution. They are generally limited to two tasks only and most involve aligning the distributions of these two tasks. In Wu & He (2022) they propose the use of meta-learning for a transfer learning problem. The method is limited to matching the empirical distribution of dynamic source and target tasks.

**Parameter Identification.** A more general form of adaptation to dynamical systems can be identified in the early work on system identification (Åström & Eykhoff (1971)). More precisely, parameter identification refer to the estimation of unobservable parameters influencing the dynamic system considered from a sequence of observations. Most of these studies however consider only one of the two required steps for adaptation. In fact, they assume to know the law governing the process and estimate the conditional parameters (Bhat et al. (2002); Yu et al. (2017)), impose a suitable inductive bias to guide the learning process (Sanchez-Gonzalez et al. (2018)) or use a hybrid approach to learn a residual of an imperfect but known system (Ajay et al. (2019)).

**Optimization-Based Meta-Learning.** Meta-learning is concerned with estimating both of the required adaptation steps from data alone. Optimization-based methods do this by performing an adaptation step on the learner itself (Ravi & Larochelle (2016)). Of these, Gradient-Based Meta-Learning is a particular case where the adaptation is performed by a gradient descend step on the parameters of the learner (Finn et al. (2017)). This family of methods have been shown to be universal function approximators (Finn & Levine (2017)). One shortcoming of these methods is that they can only be used on problems where the architecture of the function used to infer the gradient is the same as the adapted learner. Indeed, the only works considering a form of transfer assume a shift in the input space (Bennequin et al. (2021); Du et al. (2020); Jiang et al. (2022)). In contrast to past work, our method can be applied in the case of the adaptation data being of a different nature with respect to the final task.

**Model-Based Meta-Learning.** A second approach to meta-learning is to learn a model to output directly the adapted learner for the task. For problems requiring a specific adaptation strategy, in fact, the two steps can be coupled together and learned with a unique model (Xu et al. (2019); Li et al. (2018)). More general is, instead, the use of a parallel neural network, conditioned on the adaptation data, to output directly the parameters of the adapted learner. HyperNetworks are one of the most commonly used methods for this (Ha et al. (2016)). This has been done using amortized inference (Gordon et al. (2018)), gradient-based (Rusu et al. (2018); Munkhdalai & Yu (2017)) or outputting a conditional to the learner (Kirchmeyer et al. (2022)). Similarly to our work, (Xian et al. (2021)) have used HyperNetworks to estimate the unobservable properties of dynamical systems. In contrast, they require the use of a feature extractor when dealing with high-dimensional inputs. Overall, these black-box approaches can be particularly expressive but suffer generalization performances and are subject to statistical noise. Another line of work is the Neural Process Family (Garnelo et al. (2018), Kim et al. (2019)). In contrast to HyperNetworks, they generate functions by conditioning and also incorporate uncertainty through a variational approach. Wang et al. (2021) considers a model-based meta-learning framework where they learn to transfer between different tasks from the NLP domain. In contrast to our work, they use the Fisher Information Matrix of their base learner to define the task representation.

## 3 Preliminaries

We situate transfer learning as a learning-to-learn problem. As a necessary preamble, we review the general formulation of meta-learning and describe two specific instances of it as GBML and HyperNetworks respectively.

### 3.1 Meta-Learning

Meta-Learning describes a family of algorithms that are designed for *learning-to-learn*. Given a space of *tasks*, a meta-learner utilizes previous knowledge to efficiently learn new tasks using only a limited number of data samples. A task can equivalently be seen as a dataset or a function. Formally, we define a task $\mathcal{T}_f \subseteq \mathbb{X} \times \mathbb{Y}$ as a collection of input-output pairs defined by an underlying, unknown function $f : \mathbb{X} \to \mathbb{Y}$. In other words, $\mathcal{T}_f = \{(x, f(x)) | x \in \mathbb{X}\}$. From this definition, we can denote the *space of tasks* over a function space $\mathcal{F}$ as $\mathcal{T}_{\mathcal{F}}$. In fact, the function $f$ uniquely identifies the task $\mathcal{T}_f$. Throughout the paper we drop the subscript $f$ whenever the tasks functional dependence on $f$ is not of importance.

In the standard supervised learning setting, we aim to learn a function $f_\psi$ with parameters $\psi \in \Psi \subseteq \mathbb{R}^d$ that approximates a function $f$ through a supervised loss $\mathcal{L}(\mathcal{T}_f, \psi)$. The purpose of meta-learning is, instead, to find a set of optimal parameters $\psi$ from only a small dataset $\mathcal{D}_f \sim \mathcal{T}_f$. From the notation defined above, the meta-learning methodology can be formalized as learning a parameterized *update function* $\mathcal{M}_\phi : \mathcal{T}_{\mathcal{F}} \times \Theta \to \Psi$ that maps a single task $\mathcal{T}_f$ and some prior $\theta \in \Theta$ to the updated optimal parameters $\psi$ of $f_\psi$. The optimization problem can then be stated as:

$$\min_{\phi, \theta} \mathbb{E}_{\mathcal{T}_f} \left[ \mathcal{L}(\mathcal{D}_f^Q, \psi) \right] \quad \text{s.t.} \quad \psi = \mathcal{M}_\phi(\mathcal{D}_f^S, \theta) \tag{1}$$

Here, $\mathcal{D}_f^S, \mathcal{D}_f^Q$ refer to a *support* and *query* set that are sampled without replacement from task $\mathcal{T}_f$.

## 3.2 GRADIENT-BASED META-LEARNING

The framework described above is general and fits a variety of meta-learning methodologies. These methodologies mainly differ in how they implement the update function $\mathcal{M}$. In particular, GBML uses as a prior another set of parameters i.e. $\theta \in \Theta \subseteq \mathbb{R}^d$. This common set of parameters is used as the initialization among tasks such that any task can be learnt by only a few gradient-steps on a limited number of samples. In the general case, the update function can be expressed as:

$$\mathcal{M}_\phi(\mathcal{D}_f^S, \theta) = \theta - M\nabla_\theta \mathcal{L}(\mathcal{D}_f^S, \theta) \tag{2}$$

Here, $M \in \mathbb{R}^{d \times d}$ is a learnable *preconditioning* matrix which facilitates the gradient descent. In particular, we have $\phi = M$ in this case. For example, a diagonal preconditioning corresponds to learnable learning rates used in Meta-SGD (Li et al. (2017)) while a full-rank matrix corresponds to Meta-Curvature (Park & Oliva (2019)). Other forms of preconditioning have been studied in (Lee & Choi (2018); Flennerhag et al. (2019)).

## 3.3 HYPERNETWORKS

GBML incorporates the inductive bias in that adaptation to a new task necessarily implies an optimization procedure. From the general formulation expressed in Equation 1, this does not necessarily have to be the case. An alternative approach is to consider the map $\mathcal{M}_\phi : \mathcal{D} \times \Theta \rightarrow \Psi$ directly as a parameterized neural network. As such, the network takes as input the task and possibly some additional parameters and outputs the task-adapted parameters $\psi$ directly. To this end, $\mathcal{M}$ constitutes a *HyperNetwork* (Ha et al. (2016)) which are networks whose output are the weights of another neural network denoted as the *main network*. This formulation is also consistent with recurrent-based meta-learners that implement a learning algorithm through a recurrent neural network (RNN) (Santoro et al. (2016)).

The update function $\mathcal{M}$, in this case, can be formulated with a general auto-encoder structure i.e. $\mathcal{M}_\phi = \mathcal{M}_{\phi_2}^D \circ \mathcal{M}_{\phi_1}^E$. HyperNetworks take as input the support dataset $\mathcal{D}_f^S \sim \mathcal{T}_f$ of a task by encoding the $N$ input-output pairs $(x_i^S, y_i^S)_{i=1}^N$ using a non-linear function $h_{\phi_1}$. To ensure permutation-invariance, an appropriate aggregation function $\Sigma$ can be used. The latent representation $z \in \mathbb{R}^k$ of the task can then be formulated as:

$$z = \mathcal{M}_{\phi_1}^E(\mathcal{D}_f^S) = \Sigma(\{h_{\phi_1}(x_i^S, y_i^S)\}_{i=1}^N) \tag{3}$$

This representation is then passed through the decoder $\mathcal{M}^D$ to output the parameters $\psi$ of the main network $g_\psi$.

## 4 META-TRANSFER

Transfer-Learning is concerned with learning a specific target task, given knowledge of a source task. To achieve this, one can utilize some inductive bias e.g. the assumption that representations learnt in the source task can in turn be useful to learn the target task. We now ask the question if this inductive bias for transfer can be learnt as well. Instead of a single source and target, we consider a joint distribution over pairs of tasks and aim to learn to *meta-transfer*. In the meta-learning formalism we have presented up until now, we have made the assumption that the support and query set are samples from the same task w.r.t a functional $f$ i.e. $\mathcal{D}_f^S, \mathcal{D}_f^Q \sim \mathcal{T}_f$. To address the problem of meta-transfer, we extend the meta-learning formulation by considering support and query to be defined over *different* function spaces . In this respect, we have $\mathcal{D}_f^S \sim \mathcal{T}_f$ and $\mathcal{D}_g^Q \sim \mathcal{T}_g$ with $f \neq g$. To learn an efficient adaptation on the query set, there needs to be an explicit relationship between support and query. We define this as an unknown map $T$ from function to function such that $g = T(f)$. In particular, when $T = I$ we fall back to the case of standard meta-learning. On the other hand, the meta-learner needs to learn this functional dependency as well if $T \neq I$. In the next section we describe the proposed method to achieve this.

### 4.1 METHOD

When the two functions $f$ and $g$ require a different architecture for their parametric approximation, standard GBML methods cannot be used. In this section we describe an extension of GBML to

handle such cases. In particular, we propose to approximate $T$ by learning a map from function space to function space. Let $f_\theta$ be a parameterized neural network and $\mathcal{D}_f^S = (x_i^S, y_i^S)_{i=1}^N$ be the support data sampled $\mathcal{T}_f$. We argue that a good representation of the function $f$ is a function itself. In similar notation as in Section 3.3, we want to construct an encoder $\mathcal{M}^E$ of the support set. We attain this by defining $\mathcal{M}^E$ as the gradients of $f_\theta$ w.r.t. to a chosen loss function $\mathcal{L}$ on the support. However, representing a function through its parameters bears with it a problem of dimensionality. Since the parameters $\theta' \in \mathbb{R}^d$ are of a neural network, they will possibly belong to a very high-dimensional space. To solve this, we adhere to only optimizing over a subspace of the full parameter space.

In practice, there are two ways of achieving this. The first is to modulate the gradients through a conditioning variable $z$ that is concatenated with the input. The second approach is to use a hypernetwork $h_{\phi_1} : \mathcal{Z} \to \Theta$ from a low-dimensional latent-space $\mathcal{Z} \subseteq \mathbb{R}^k$ to the parameter space of the learner $f$. More specifically, the first approach was explored in the meta-learning literature as CAVIA (Zintgraf et al. (2019)) while the second approach corresponds to the method employed in LEO (Rusu et al. (2018)). This last method can be further divided into two variations based on the implementation of the function $h_{\phi_1}$. This can be, in fact, either a linear or a non-linear neural network. The linear approach would effectively imply a linear projection of the gradients to a low-dimensional subspace.

$\mathcal{M}^E$ gives a task representation $z$ in a similar vein as equation 3. This representation of the function $f$ can thus be extracted by means of $\mathcal{M}^E$ using one of these three methods summarized below:

- **Context**: We concatenate the input to learner $f$ with a parameter $z \in \mathbb{R}^k$ to modulate the output. Thus:

$$\mathcal{M}^E(\mathcal{D}_f, \xi) = \mathbb{E}_{x,y \sim \mathcal{D}_f} \left[ \nabla_z \mathcal{L}(f_\theta(x, z), y) \right], \quad \xi = [z, \theta] \tag{4}$$

- **Non-Linear**: The task-adapted parameters are modulated through a latent parameter $z \in \mathbb{R}^k$. This is achieved through the use of a hypernetwork $h_{\phi_1}$ mapping from the latent space to parameter space:

$$\mathcal{M}^E(\mathcal{D}_f, \xi) = \mathbb{E}_{x,y \sim \mathcal{D}_f} [\nabla_z \mathcal{L}(f_{h_{\phi_1}(z)}(x), y)], \quad \xi = [z, \phi_1] \tag{5}$$

- **Linear**: In the case of linearity we rewrite the hypernetwork as $V = h_{\phi_1}$. The matrix $V \in \mathbb{R}^{k \times d}$ is employed to linearly project the gradients to a lower-dimensional subspace. We thus calculate the $k$ directional derivatives w.r.t $[\mathbf{v_1}, \ldots, \mathbf{v_k}]^T = V$:

$$\mathcal{M}^E(\mathcal{D}_f, \xi) = \mathbb{E}_{x,y \sim \mathcal{D}_f} [V \nabla_\theta \mathcal{L}(f_\theta(x), y)], \quad \xi = [V, \theta] \tag{6}$$

The representation of the support task can then be used to estimate an optimal learner on the related downstream task $\mathcal{T}_g$. This, in turn, requires decoding this representation to a set of parameters that are optimal on the query dataset $\mathcal{D}_g^Q$. For this, we employ a decoder network $\mathcal{M}^d$ on $z$ that outputs the parameters $\psi \in \Psi$ of a neural network $g_\psi$. In summary, our update function is defined as

$$\mathcal{M}(\mathcal{D}_f) = \mathcal{M}^D(\mathcal{M}^E(\mathcal{D}_f, \xi)) \tag{7}$$

We train the model end-to-end by optimizing the loss of $g_\psi$ on the query set $\mathcal{D}_g^Q$ on every training task through gradient-descent. The final objective of the model can thus be formulated as:

$$\min_\xi \mathbb{E}_{\mathcal{T}_g \sim \mathcal{T}_\mathcal{G}, \mathcal{T}_f \sim \mathcal{T}_\mathcal{F}} \left[ \mathcal{L}(\mathcal{D}_g, \psi) \right] \quad s.t. \quad \psi = \mathcal{M}^E(\mathcal{D}_f, \xi) \tag{8}$$

## 5 EXPERIMENTS

We validate our approach on three different classes of problems involving regression, dynamics prediction and imitation learning. For each of these problems, we evaluate the ability to transfer in the presence of increasing amount of noise in the support. We further assess the representational capability of our model by learning to predict the ground-truth task parameters from the learnt representation. Finally, we evaluate our model's capacity to learn high-dimensional representations by considering a maze environment in which the ground-truth task parameters are potentially high-dimensional and unknown.

## 5.1 BASELINES

We consider different encoders $\mathcal{M}_E$ as our baselines. In past work (Xian et al. (2021), Garnelo et al. (2018)), the mean pooling function is used. For comparison, we also consider the MAX operation. For context, mean pooling would correspond to DeepSets, (Zaheer et al. (2017)) while max pooling would correspond to PointNet (Qi et al. (2017)). Furthermore, we consider an encoder based on the transformer architecture (Vaswani et al. (2017)) in which we find a weighted average of all the samples in the support where the weights are the pairwise dot-products between encodings of the data points. We refer to the different aggregation schemes as pooling methods in our experiments. Lastly, we consider a deep-kernel architecture based on MetaFun (Xu et al. (2020)). Here, points of the query are directly compared to points from the support based on a learnt kernel-function. As such, we do not achieve an intermediate latent representation $z$ of our task but rather we directly output a function $f$ defined as $f = \sum_{i=1}^{N} k(\cdot, x_i^S) r(x_i^S, y_i^S)$ with $k, r$ as parameterized neural networks. For the gradient-based encoders, we consider the three different methods outlined in Equations 4-6. To enable a fair comparison between the methods, we employ the same decoder network for all models except MetaFun. Implementation details can be found in the Appendix. We use the same subspace dimension $k$ for all models and let $k$ equal the true dimension of the task-space when known as we found this is sufficient to achieve a good performance.

## 5.2 SYNTHETIC REGRESSION TASK

As a proof-of-concept, we consider the problem of transfer learning between two sinusoidal functions. We consider the support data to be regression tasks drawn from $y = A\cos(x + b)$ with $A \in [0.1, 5.0]$ and $b \in [-\pi, \pi]$. The corresponding query tasks are constructed from a sine wave with the same amplitude $A$ and phase $b$ as the support task. The task is then to essentially extract the task-parameters $A, b$ from the support to learn the query task. In this experiment, we highlight our methods robustness to white noise in the support. The results can be seen in Table 1. As the amount of noise increases, the performance of gradient-based encoders remains constant while other pooling methods eventually become unstable. Figure 2 shows qualitative results. The figure shows the sinusoids found through the linear-projection of the gradient compared to the ground-truth and MEAN and MAX pooling. We evaluate the models on 100 different samples of noise and plot the mean and standard deviation. The results show the robustness of our method even up to noise with a standard deviation of 4.0.

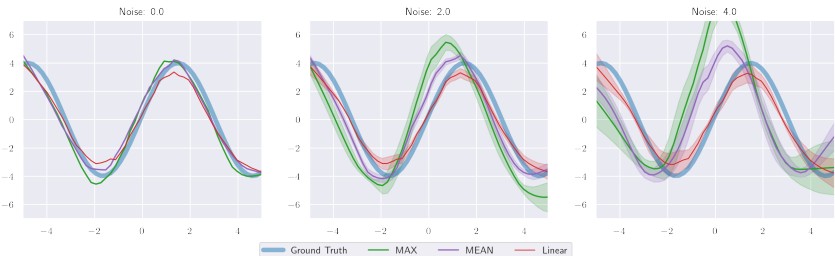

Figure 2: Sinusoid regression for different levels of noise. Using the gradients as an encoder stays invariant to the amount of noise in the support data.

## 5.3 DOUBLE INVERTED PENDULUM

For our next experiment we consider the more complex, real-world scenario of learning the dynamics of a physical system. The system we consider is the double inverted pendulum. We hypothesize that from observing the dynamics of a *single* pendulum, one can infer the dynamics of a *double* pendulum that shares the same global physical parameters and object properties as the single pendulum. The single and double pendulum are simulated using the Mujoco environment (Brockman et al. (2016)) and involve a pendulum attached to a cartpole which can move left and right. Our support task thus consists of state-action-state triples drawn from the single pendulum. Concretely, we have $\mathbb{X}^S = \mathcal{S} \times \mathcal{A}$ and $\mathbb{Y}^S = \mathcal{S}$ with $\mathcal{S}$ defined as the state of the pendulum denoted by its position, velocity, angle and angular velocity, $(x, \dot{x}, \theta, \dot{\theta})$ and $\mathcal{A} = [-3, 3]$ representing the force applied to

| | Model | $\sigma = 0.0$ | $\sigma = 0.5$ | $\sigma = 1.0$ | $\sigma = 1.5$ |
|---|---|---|---|---|---|
| GRADIENT | - CONTEXT | $0.23_{\pm 0.04}$ | $0.35_{\pm 0.07}$ | $0.49_{\pm 0.10}$ | $0.61_{\pm 0.10}$ |
| | - NON-LINEAR | $0.41_{\pm 0.10}$ | $0.46_{\pm 0.05}$ | $0.57_{\pm 0.09}$ | $0.69_{\pm 0.08}$ |
| | - LINEAR | $0.31_{\pm 0.05}$ | $0.36_{\pm 0.09}$ | $0.43_{\pm 0.08}$ | $\mathbf{0.51}_{\pm \mathbf{0.05}}$ |
| POOLING | - MEAN | $0.24_{\pm 0.03}$ | $0.25_{\pm 0.03}$ | $\mathbf{0.44}_{\pm \mathbf{0.02}}$ | $0.86_{\pm 0.11}$ |
| | - MAX | $\mathbf{0.13}_{\pm \mathbf{0.06}}$ | $\mathbf{0.22}_{\pm \mathbf{0.04}}$ | $0.61_{\pm 0.15}$ | $1.31_{\pm 0.24}$ |
| | - TRANSFORMER | $0.26_{\pm 0.04}$ | $0.80_{\pm 0.02}$ | $1.34_{\pm 0.05}$ | $2.30_{\pm 0.18}$ |
| | DEEP KERNEL | $0.75_{\pm 0.25}$ | $0.87_{\pm 0.04}$ | $1.12_{\pm 0.03}$ | $1.58_{\pm 0.09}$ |

Table 1: Results for the *cosine-sine* experiment. The top three rows compares different methods of encoding the gradients based on LINEAR, NON-LINEAR and CONTEXT methods. We compare all models against different standard deviations of noise in the support ($\sigma$). From the results, gradient-based encodings are more robust as the amount of noise increases.

the base cart. The corresponding query task is forward dynamics prediction on the double pendulum. Hence, $\mathbb{X}^Q$ and $\mathbb{Y}^Q$ are state-action triples as before. We experiment with varying degrees of noise to confirm our models robustness properties. For the support data, we add noise $\tilde{y}^S = y^S + \varepsilon$ with $\varepsilon \sim \mathcal{N}(0, \sigma^2)$. We experiment with various $\sigma \in [0.4, 1.0, 2.0, 3.0]$ respectively. The results for the forward dynamics prediction are shown in the Appendix in Table 4. To further confirm our findings on noise robustness, we perform an additional experiment where we instead consider the task of regression to the physical parameters of the single pendulum. We plot the MSE to the physical parameters (pendulum length and gravity) against the standard deviation of noise in the support. We evaluate each model 100 times and plot the mean and standard deviation. The results are shown in Figure 3. We can note similar performance across all models for low levels of noise. As the amount of noise increases however, the MAX aggregation quickly explodes. The MEAN aggregator is more robust but similarly is also not unaffected by noise in the support. In this experiment, we also note that methods based on non-linear projections of gradients such as CONTEXT and NON-LINEAR show better performance than the linear projection.

## 5.4 IMITATION LEARNING

| | Model | $\sigma = 0.0$ | $\sigma = 1.0$ | $\sigma = 2.0$ | $\sigma = 4.0$ |
|---|---|---|---|---|---|
| GRADIENT | - CONTEXT | $4.075_{\pm 0.214}$ | $4.220_{\pm 0.154}$ | $\mathbf{4.321}_{\pm \mathbf{0.262}}$ | $\mathbf{4.361}_{\pm \mathbf{0.152}}$ |
| | - NON-LINEAR | $\mathbf{4.255}_{\pm \mathbf{0.010}}$ | $\mathbf{4.277}_{\pm \mathbf{0.144}}$ | $4.295_{\pm 0.121}$ | $4.096_{\pm 0.027}$ |
| | - LINEAR | $3.999_{\pm 0.112}$ | $4.062_{\pm 0.108}$ | $4.173_{\pm 0.106}$ | $4.052_{\pm 0.159}$ |
| POOLING | - MEAN | $3.929_{\pm 0.061}$ | $3.930_{\pm 0.064}$ | $3.955_{\pm 0.344}$ | $3.370_{\pm 0.538}$ |
| | - MAX | $4.108_{\pm 0.155}$ | $4.027_{\pm 0.157}$ | $3.958_{\pm 0.325}$ | $3.139_{\pm 1.060}$ |
| | - TRANSFORMER | $3.947_{\pm 0.102}$ | $4.199_{\pm 0.223}$ | $4.037_{\pm 0.104}$ | $-30.564_{\pm 29.494}$ |
| | DEEP KERNEL | $0.086_{\pm 0.086}$ | $-0.001_{\pm 0.169}$ | $0.099_{\pm 0.183}$ | $-0.192_{\pm 0.134}$ |

Table 2: Final reward for the imitation learning experiment. Higher is better.

We conduct an experiment to probe our models ability to infer an optimal policy for a given MDP given only a few observations from the environment. We consider a modification of the Mujoco Ant environment where we vary the length of the legs as our testing ground (see Figure 4 and Appendix).

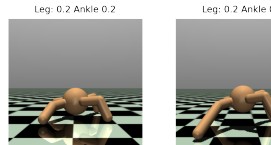 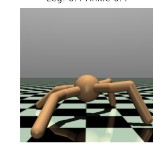 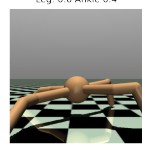 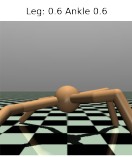

Figure 4: Visualization of the modified Ant environment. We vary the length of the upper and bottom parts of the leg to create different conditions for the control task.

The goal of this experiment is to test the models ability to transfer between dynamics data and a policy estimation. Here we are not interested in inferring an optimal exploration strategy but rather assume that the dynamics data contains all the relevant information to infer the optimal behavior for the agent. We avoid the use of a reinforcement learning loss as it has been shown that common automatic-differentiation tools can't compute the second derivative of Monte-Carlo expectations (Rothfuss et al. (2018); Foerster et al. (2018)). The target loss is thus a behavioral cloning of a second policy trained with privileged information. We train PPO (Schulman et al. (2017)), conditioned on the ground-truth physical parameters. Our query task is then from a given state to infer the action given by the optimal policy. We condition our meta-imitation

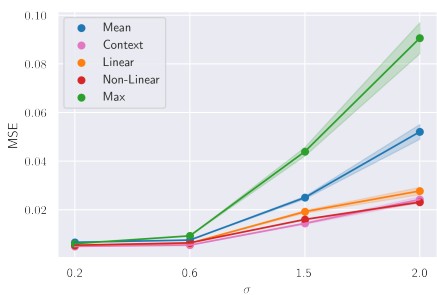

Figure 3: MSE of regression to the ground-truth task parameters from a finite set of interactions with a single pendulum.

learner by considering a trajectory $(s_t, a_t, s_{t+1})_{t=1}^T$ gathered from a random policy as our support data. We train for 100 epochs on a dataset of 100 different tasks with $T = 30$ number of points in the support. Each task is constructed by sampling an upper-leg length and ankle length from $\mathcal{U}(0.2, 0.6)$. We experiment with inputting varying amount of noise in the support data during testing to validate our models robustness to noise. We show the final reward after one episode for the methods in Table 2. We note that even with noise infused on the support set, our method achieves a consistent high reward while for our baselines they slowly decline. With random noise added with $\sigma = 8.0$, the transformer-based architecture achieves a large negative reward while the deep-kernel method failed to transfer in all cases.

### 5.5 MAZE EXPERIMENT

We implement a simple maze environment defined over a $2 \times N$ grid. We define the starting point of our agent in upper left corner and the goal is to reach the lower right corner as efficiently as possible. To generate different mazes, we randomly place obstacles on the grid in such a way that the maze is always solvable (see Appendix for further details). The optimal policy is in this case deterministic and found through a depth-first search. For the support data we let $\mathbb{X}^S \subset \mathbb{Z}^2$ be positions of the grid and $\mathbb{Y}^S = \{0, 1\}$ be binary variable indicating if a position in the grid is occupied or not. We experiment with varying the width $N$ of the maze by letting $N \in [6, 12, 24]$.

The results can be seen in Table 3. In contrast to previous experiments, the advantage of deep kernels become apparent in this task. Furthermore, the linear method achieves a significant performance gain over the other methods.

## 6 DISCUSSION

### 6.1 THE ADVANTAGE OF GRADIENT-BASED ENCODERS

The empirical gradient of a function $\nabla \mathcal{L}$ is an unbiased estimator of the true gradient. As such, it possesses a form of consistency as it converges in distribution to the true expected gradient of the loss over the entire data-space. This, results in statistical advantages over black-box representations.

|  | Model | MAZE-SIZE $= 6$ | MAZE-SIZE $= 12$ | MAZE-SIZE $= 24$ |
|---|---|---|---|---|
| GRADIENT | - CONTEXT | $-0.237_{\pm 0.022}$ | $-0.368_{\pm 0.000}$ | $-0.435_{\pm 0.000}$ |
| | - NON-LINEAR | $0.000_{\pm 0.000}$ | $-0.348_{\pm 0.037}$ | $-0.415_{\pm 0.015}$ |
| | - LINEAR | $0.000_{\pm 0.000}$ | $\mathbf{-0.036_{\pm 0.029}}$ | $\mathbf{-0.355_{\pm 0.041}}$ |
| POOLING | - MEAN | $0.000_{\pm 0.000}$ | $-0.291_{\pm 0.020}$ | $-0.396_{\pm 0.003}$ |
| | - MAX | $0.000_{\pm 0.000}$ | $-0.156_{\pm 0.093}$ | $-0.386_{\pm 0.017}$ |
| | - TRANSFORMER | $-0.028_{\pm 0.039}$ | $-0.275_{\pm 0.029}$ | $-0.400_{\pm 0.025}$ |
| | DEEP KERNEL | $0.000_{\pm 0.000}$ | $-0.257_{\pm 0.010}$ | $-0.428_{\pm 0.010}$ |

Table 3: Final reward as measured by the normalized distance to goal position for the maze experiment. The results show that a linear projection can efficiently encode and transfer the information gathered from the exploration phase while other gradient-based methods fail compared to the baselines.

One being resilience in overfitting and better generalization to out-of-domain tasks, as shown in Finn & Levine (2017). Another being robustness to white noise in both the input and the output of the support data, as empirically shown in this paper. This robustness to noise makes the use of gradient as a representation appealing for real-world applications where noisy observations and faulty labeling is often unavoidable. Moreover, the gradient carries a semantic meaning in that it will always point in the direction of steepest descent given the data.

## 6.2 LIMITATIONS

Utilizing gradient-information in an end-to-end manner requires computing the second order derivative during training. This can be prohibitively expensive in large-scale experiments. Using first-order methods such as Reptile (Nichol et al. (2018)) could potentially be incorporated into our method to alleviate this concern. A second limitation is the dimensionality of the parameter space. When dealing with large networks this might cause memory-related issues. Lastly, even though theoretically sound, second-order Monte Carlo expectations cannot currently be handled by automatic differentiation tools. This prevents the use of any GBML method with loss functions involving these expectations e.g. reinforcement learning losses.

## 7 CONCLUSION

In this paper we have proposed a general framework for transferring knowledge from one task to another from an adaptation perspective. To this end, we have described a family of methods based on the intersection between GBML and model-based techniques. Furthermore, we have explored the use of gradients as a task representation and the advantages of such with respect to other representations. We have empirically demonstrated the advantages of this representation on a number of experiments. Such advantages are especially noteworthy in case of statistical errors in the adaptation data like the presence of white noise. These results not only reinforce the advantages of gradient-based meta-learning but exemplify how the same methodology can be extended to novel problems where gradient-based methods have previously been unexplored.

As a future line of work, one can further connect our method to the neural process family by explicitly incorporating uncertainty measures into the encoder. This can for example be achieved by imbuing MAML in a probabilistic framework as done in Finn et al. (2018). Another possible extension is the use of hierarchical GBML methods to handle complex distributions of tasks.

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

# A APPENDIX

## A.1 IMPLEMENTATION DETAILS

In all of our experiments, we let the implicit representation learner $f_\theta$ be a 3-layer MLP with 40 hidden units. For the non-linear method based on LEO Rusu et al. (2018), we implement the hyper-network as a 2-layer MLP with 128 hidden units.

For the pooling methods, we let $h$ be an MLP with 3 hidden layers of width 128 and a final output layer of size $k$. For the transformer, we first encode each input separately with $h$. We then implement the query, key and value networks as linear layers before we calculate the output as SOFTMAX$(QK^T)V$. For METAFUN, we encode each point in the support and query through a 2-layer MLP into a 128-dim vector. We then utilize an RBF kernel with temperature $\tau = 1.0$. The outputs are multiplied by the outputs of a network $r$ that is implemented as another 2-layer MLP. We employ a final network $w$ on the outputs to find $y^Q = w\left(\sum_{i=1}^N k(x^Q, x_i^S) r(x_i^S, y_i^S)\right)$

We use a learning-rate of $5e - 4$ and train for 100 epochs. All results presented are averaged over three different random seeds with the mean and standard deviation calculated on different held-out test sets. For all experiments we use a 2-layer MLP for the decoder network $\mathcal{M}^D$.

## A.2 ALGORITHM

---
**Algorithm 1** Meta-Transfer
---
**Require:** $p(\mathcal{T}_\mathcal{F}), p(\mathcal{T}_\mathcal{G})$: distributions over tasks, randomly initialize $\xi$
    **while** not done **do**
        sample task $T_f \sim p(\mathcal{T}_\mathcal{F}), T_g \sim p(\mathcal{T}_\mathcal{G})$
        sample batch of data-points $\mathcal{D}_f \sim T_f, \mathcal{D}_g \sim T_g$
        $z = \mathcal{M}^E(\mathcal{D}_f, \xi)$                                 ▷ According to Equation 4, 5, 6
        $\psi = \mathcal{M}^D(z)$
        update $\xi \leftarrow \xi - \alpha\nabla_\xi \mathcal{L}_{T_g}(\mathcal{D}_g, \psi)$
    **end while**
---

## A.3 ADDITIONAL PENDULUM EXPERIMENTS

| | Model | $\sigma = 0.0$ | $\sigma = 0.4$ | $\sigma = 1.0$ | $\sigma = 2.0$ | $\sigma = 3.0$ |
|---|---|---|---|---|---|---|
| GRADIENT | - CONTEXT | $0.33_{\pm 0.01}$ | $0.33_{\pm 0.02}$ | $0.33_{\pm 0.02}$ | $0.34_{\pm 0.01}$ | $0.35_{\pm 0.01}$ |
| | - NON-LINEAR | $0.33_{\pm 0.01}$ | $0.32_{\pm 0.02}$ | $0.32_{\pm 0.02}$ | $0.33_{\pm 0.02}$ | $0.38_{\pm 0.07}$ |
| | - LINEAR | $0.32_{\pm 0.02}$ | $0.33_{\pm 0.03}$ | $0.33_{\pm 0.03}$ | $0.35_{\pm 0.01}$ | $0.41_{\pm 0.05}$ |
| POOLING | - MEAN | $0.33_{\pm 0.01}$ | $0.33_{\pm 0.01}$ | $0.35_{\pm 0.01}$ | $0.42_{\pm 0.03}$ | $0.60_{\pm 0.14}$ |
| | - MAX | $0.31_{\pm 0.02}$ | $0.35_{\pm 0.02}$ | $0.40_{\pm 0.07}$ | $0.57_{\pm 0.22}$ | $1.19_{\pm 0.84}$ |
| | - TRANSFORMER | $0.30_{\pm 0.00}$ | $0.32_{\pm 0.02}$ | $0.34_{\pm 0.02}$ | $0.73_{\pm 0.39}$ | $3.45_{\pm 1.27}$ |
| | DEEP KERNEL | $1.35_{\pm 0.05}$ | $1.36_{\pm 0.03}$ | $1.67_{\pm 0.06}$ | $2.56_{\pm 0.54}$ | $3.14_{\pm 0.98}$ |
| | $\mathbb{E}[(x_{t+1} - x_t)^2]$ | $3.23_{\pm 0.00}$ | | | | |

Table 4: Results for double pendulum experiment

For reference we also compute the average distance between consecutive points. This would correspond to always predicting the current state, thus computing $\mathbb{E}[(x_{t+1} - x_t)^2]$.

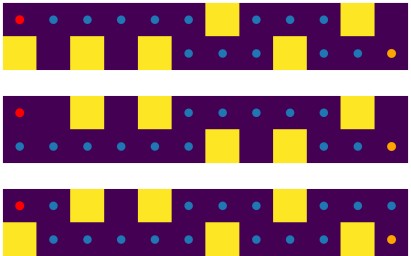

Figure 5: Three examples of the maze environment. The start and goal position are defined by the red and orange marker respectively.

## A.4 Ant Experiment Details

In the environment, the agent consists of a torso and four legs. Each leg is composed of two links (upper-leg, ankle) joint together. To generate different tasks, we vary the length of the upper-leg and the ankle for all the legs. To retain symmetry, for each of the four legs we consider the same adjustment and thus leaving us with a two degree-of-freedom change between the tasks. The goal of the agent is to walk as far as possible in the $x$-direction while maintaining a certain stability.

## A.5 Maze Experiment Details

We define the maze over a $2 \times N$ grid. From every other position on the top row of the maze ($\frac{N}{2}$ positions) we sample an obstacle with probability $p = 0.5$. We then define the obstacles on the bottom row as the complement of the top row. Three examples can be seen in Figure 5

