# OpenReview forum: "Gradient-Based Transfer Learning"
_ICLR.cc/2023/Conference — Submitted to ICLR 2023_

### Official Review · Reviewer_6CPS · 2022-10-24

**Confidence:** 3
**Correctness:** 3
**Technical Novelty And Significance:** 2
**Empirical Novelty And Significance:** 3
**Recommendation:** 3

**Clarity, Quality, Novelty And Reproducibility:**

The paper is clearly written. As far as I know this method is novel, although I can't guarantee that using gradient information has not been used in prior work on model-based meta-learning.

The source code as been provided in the supplementary material. Although I have not checked the code in details, some information about data collection for some environments (e.g. pendulum) is missing. Many hyperparameters have not been included in the Appendix (e.g., the number of directions for the "linear" method).

**Details Of Ethics Concerns:**

No ethics concerns

**Strength And Weaknesses:**

**Strengths**: The presentation of the theory is for the most part precise. The empirical evaluation is extensive, over multiple environments with varying degrees of complexity. The comparison with standard model-based meta-learning methods as the encoder $M^{E}$ is appreciated.

**Weakness**:
 1. What is the connection between your proposed method (as introduced in Section 4.1) and GBML, as presented in Section 3.2? It seems that the proposed method does not use any adaptation through and optimization procedure as in GBML, but is closer to model-based meta-learning, where the encoder $M^{E}$ happens to use gradient information. As such, it is incorrect to call the proposed method GBML (as in Section 7, "*a family of method at the intersection between GBML and model-based techniques*", when the proposed method is purely model-based).
 2. Although most of the theoretical part is precise, it is crucially missing some details when it comes to the proposed method. In particular, there is no detail as to the point at which the gradient information used in $M^{E}$ is evaluated. For example in Equation 4, $\mathbb{E}[\nabla_{z}L(f(x, z), y)]$, at what point $z$ is this gradient evaluated? Is $z$ a learned meta-parameter? This is crucially missing from the description of the method; for example, if $z$ is meta-learned, then you should explicitly write down the meta-learning objective in the paper.
 3. Although the experiment on next-state prediction for pendulum/double pendulum is interesting, in practice what is the temporal resolution of the data? I feels like this problem might be too simple if the temporal resolution is very high.

**Summary Of The Paper:**

The authors proposed a novel model-based meta-learning method, specifically applied to the problem of transfer learning, where support and query data come from two separate (but related) tasks. This formulation allows flexibility in terms of models that can be applied, in that the model used for support examples need not be the same as the one for query examples. This method is based on gradient information on the support loss, and conditions the parameters of the model on the query examples. This method has been compared to other model-based meta-learning methods, on various environments.

**Summary Of The Review:**

There are crucially missing details in the current submission regarding the proposed method, and some confusion regarding how to frame this method. That's why I am currently recommending rejection, but I am willing to increase my score if these concerns are properly addressed.

---

> ### Author Response · Authors · 2022-11-16
> **Reply to reviewer 6CPS**
>
> We thank to reviewer for raising some important questions on the paper and we would like to address these comments below.
>
> **What is the connection between your proposed method (as introduced in Section 4.1) and GBML, as presented in Section 3.2? It seems that the proposed method does not use any adaptation through and optimization procedure as in GBML, but is closer to model-based meta-learning, where the encoder $M^E$  happens to use gradient information. As such, it is incorrect to call the proposed method GBML (as in Section 7, "a family of method at the intersection between GBML and model-based techniques", when the proposed method is purely model-based).**
>
> The reviewer raises an important concern that we wish to expand further upon. One of the motivations of this work is to show the similarity between the two seemingly different approaches.
>
> Gradient-based meta-learning can be expressed as an encoder-decoder architecture
> $\theta’ = \mathcal{M}^D \circ  \mathcal{M}^E (D_S)$ where the encoder $\mathcal{M}^E$ is the gradient $\theta - \nabla_{\theta}\mathcal{L}$ and decoder $\mathcal{M}^D$ is the identity or even a linear function $Mx$ as in for example meta-curvature [1].
>
> Looking at this through a gradient-based meta-learning perspective, our method can be seen as a  generalisation by considering any arbitrary decoder $\mathcal{M}^D$.
> From a model-based meta-learning perspective, instead of using a learnt representation, we utilise the gradients of the error of the base model. We argue that this representation is more robust to noise.
>
>
> **Although most of the theoretical part is precise, it is crucially missing some details when it comes to the proposed method. In particular, there is no detail as to the point at which the gradient information used in is evaluated. For example in Equation 4, , at what point  is this gradient evaluated? Is a learned meta-parameter? This is crucially missing from the description of the method; for example, if is meta-learned, then you should explicitly write down the meta-learning objective in the paper.**
>
> The reviewer is correct in that the gradient is evaluated at a meta-learnt initialization of $z$ as is common practice in gradient-based meta learning. We’ve added pseudocode of our proposed algorithm in the Appendix.
>
>
> **Although the experiment on next-state prediction for pendulum/double pendulum is interesting, in practice what is the temporal resolution of the data? I feels like this problem might be too simple if the temporal resolution is very high.**
>
> The temporal resolution is 0.04 seconds. We’ve added a simple baseline that returns the current state as the next state prediction. As can be shown from the table, this results in a much higher error than our model and baselines.

---

### Official Review · Reviewer_9QJw · 2022-10-27

**Confidence:** 3
**Correctness:** 2
**Technical Novelty And Significance:** 1
**Empirical Novelty And Significance:** 2
**Recommendation:** 3

**Clarity, Quality, Novelty And Reproducibility:**

### Clarity
Generally speaking, I think that the clarity of the paper can be improved.
- The green arrow in Figure 1 is difficult to see.
- What is $g_\phi$ in the caption of Figure 1?
- As is convention, the best method for each experiment type should be in bold in tables in the Appendix.
- Does $\psi$ correspond to the optimal parameters? Then is $\phi$ the optimal parameters of a source task?
- I think that this paper would benefit from a more complete mathematical description of what is happening.  For example Section 4.1 mentions that the encoder is constructed as the gradients of $f_\theta$. I'm not sure what this means.


### Quality
This paper could go through a quick grammar and presentation pass.
- Minor grammatical errors.
- Minor capitalisation errors.
- Why are the $\pm$ values written as subscripts?
- The notatation $dataset \sim T_f$; do you mean a subset? $T_f$ is a set, not a distribution. I guess you are trying to say that it is a sampled subset of $T_f$. I haven't seen this notation elsewhere, but if you are introducing it, please clarify what it means in the text.

### Novelty
I think that this paper makes an interesting connection between transfer learning and meta-learning.

### Reproducibility
I believe sufficient experimental details are given, along with hyperparameters used. However, code doesn't seem to be given, unless I missed something.

**Strength And Weaknesses:**

### Strengths
-

### Weaknesses
Some things aren't clear to me:
- Section 2: Why can't transfer learning be considered to perform general adaptation? What is your definition of general adaptation?
- In Section 4, you mention that support and query sets are from the same task? My understanding is that $T_f$ contains all possible pairs of $x \in X$ and $y \in Y$ such that $f(x) = y$. So then, are you claiming that the datasets used in general meta-learning are simply just different sample sets of $T_f$, under the same $f$? I don't think this is true. My understanding is that $f$ can change, within an assumed (function) space.
- The double pendulum is a chaotic system. Transferring from a single pendulum here, although makes intuitive sense, seems to me could be difficult problem. A discussion regarding how this is addressed, either implicitly, or explicitly would have been nice to see.
- Are the experiments only comparing against itself? Discussing why this is the case (other methods aren't comparable?) is needed. Could you have checked against vanilla meta-learning as a baseline?

**Summary Of The Paper:**

This paper provides an meta-learning styled transfer learning method that essentially applies meta-learning to the case where supervised learning tasks vary w.r.t. their underlying functions. Experiments are performed over different problems (including regression, learning dynamics, and imitation learning).

**Summary Of The Review:**

My score is mainly based on some points made in the paper that seem dubious to me, and raise concerns regarding the correctness of this work. These are listed above. Perhaps I have misunderstood the paper, and thus look forward to the discussion period.

---

> ### Author Response · Authors · 2022-11-16
> **Reply to reviewer 9QJw**
>
> We understand the concerns of the reviewer and as follows is a reply to the ones raised.
>
> **Section 2: Why can't transfer learning be considered to perform general adaptation? What is your definition of general adaptation?**
>
> The standard transfer learning literature focuses on learning an optimal transfer of a model trained on one domain to another (different) domain. In this paper we describe the more general setting of transferring between a family of tasks to another family of tasks. We thus define a general adaptation as transfer between any source and target tasks (in a given distribution).**
>
> **In Section 4, you mention that support and query sets are from the same task? My understanding is that Tf contains all possible pairs of x∈X and y∈Y such that f(x)=y. So then, are you claiming that the datasets used in general meta-learning are simply just different sample sets of Tf, under the same f? I don't think this is true. My understanding is that f can change, within an assumed (function) space.**
>
> To clarify, in the paper, we are indeed claiming that in standard meta-learning, support and query are sampled from a task with the same underlying function. There can be a shift in the domain, for example black & white images in the support and colored images in the query but the underlying (labelling) function does not change. Our setting explicitly accounts for the case where the underlying function changes as well (between support and query), within an assumed function space which to the extent of our knowledge has not been considered before. [1]
> It is worth pointing out that in standard meta-learning (and in our setting as well) the function between the tasks does change within an assumed function space (for example sine waves of different amplitudes), we only consider the additional case where the function changes between support and query as well.
>
> [1] Bridging few-shot learning and adaptation: new challenges of support-query shift, Bennequin et al. 2021
>
> **The double pendulum is a chaotic system. Transferring from a single pendulum here, although makes intuitive sense, seems to me could be difficult problem. A discussion regarding how this is addressed, either implicitly, or explicitly would have been nice to see.**
>
> We understand the reviewers point of view. From the single pendulum, we find a latent representation of the parameters of the system and perform next state predictions of the double pendulum. The double pendulum is indeed a chaotic system in that a small change in the initial conditions will lead to a significantly different behaviour over a large time scale. In our experiment, we only test the performance over a small enough time scale to avoid the chaoticity of the system. The main point of the paper is showcasing our methods robustness to noise compared to the baselines on the 1-step prediction problem, as can be seen in Table 4 in the Appendix.
>
> **Are the experiments only comparing against itself? Discussing why this is the case (other methods aren't comparable?) is needed. Could you have checked against vanilla meta-learning as a baseline?**
>
> In the experiments we compare different functional representation methods such as taking the average, max and implicit methods such as MetaFun based on kernel methods. We also compare to different gradient based methods which could be considered comparing against ourselves.
> Vanilla meta-learning is a special case of our method, and would not necessarily make sense when there is a shift in the dimension of the output.

---

### Official Review · Reviewer_8eQb · 2022-10-27

**Confidence:** 3
**Correctness:** 3
**Technical Novelty And Significance:** 3
**Empirical Novelty And Significance:** 2
**Recommendation:** 5

**Clarity, Quality, Novelty And Reproducibility:**

The paper is clear and a gradient-based task representation is novel to my knowledge.

**Strength And Weaknesses:**

I’m assuming things like theta are also trained end-to-end in equation (8). If so, this should be stated more explicitly.

For the “context” variant for example, does the learned theta perform well on the source task, or does this network only produce good gradients for task representation while producing meaningless outputs?

Across all experiments, what is the performance of each method without noise? I think that is important for getting a sense of how severely noisy these settings (e.g. sigma=2.0) are.

Experimental evaluation is somewhat limited.

I think the problem setting is interesting; going beyond the i.i.d. assumption within  the source and query data is a nice direction. Though, I wouldn’t agree that “transfer learning” is the best way to describe it. I think transfer learning generally refers to the setting where you have some amount of information about the target task, whereas in this setting we directly produce a model for T_g before seeing any data from the domain.

The experiment in Figure 2 is compelling: gradient-based function representation automatically cancels out noise.

Page 4: The language around functionals is confusing. You call f and g functionals, but I think they are standard functions? You also say that T is a map from function to function, but then you do function composition between T and f, which implies that T is a function from Y to Y (where Y is the codomain of f).

**Summary Of The Paper:**

The core of the method is its function representation, which takes the expectation of a low-dimensional gradient over the source set. The paper presents three variants for this step. The decoder is a hyper network which maps the function representation to parameters.

**Summary Of The Review:**

This paper proposes an interesting idea but its empirical evaluation is not very strong.

---

> ### Author Response · Authors · 2022-11-16
> **Reply to reviewer 8eQb**
>
> We thank the reviewer for their comments and we would like to address the concerns.
>
> **I’m assuming things like theta are also trained end-to-end in equation (8). If so, this should be stated more explicitly.**
>
> The reviewer is correct, the proposed framework is trained end-to-end in a meta learning fashion. We have clarified this in the updated version of the paper.
>
> **For the “context” variant for example, does the learned theta perform well on the source task, or does this network only produce good gradients for task representation while producing meaningless outputs?**
>
> The learnt theta does not necessarily perform well on the source task, it is strictly used to find a representation for the target task. The main argument of the paper is that the gradient of the loss function of the base model can be used as a representation of the task. Moreover, the proposed loss function can be coupled with a loss on the performances of the base model as well to get good performances on the source task.
>
>
> **Across all experiments, what is the performance of each method without noise? I think that is important for getting a sense of how severely noisy these settings (e.g. sigma=2.0) are.**
>
> We agree with the reviewer that this would be a valuable insight to include in the paper. We have updated the numbers in the paper with $\sigma=0$ for the relevant experiments. We noticed that the results for the imitation learning experiment were already run with zero noise, although the table was labelled incorrectly. For the other experiments we have added more results.
>
>
> **I think the problem setting is interesting; going beyond the i.i.d. assumption within the source and query data is a nice direction. Though, I wouldn’t agree that “transfer learning” is the best way to describe it. I think transfer learning generally refers to the setting where you have some amount of information about the target task, whereas in this setting we directly produce a model for T_g before seeing any data from the domain.**
>
> We refer to transfer learning in the sense that we store knowledge from solving one task and use that knowledge to solve a different but related task. In usual transfer learning, one would solve the source task, and use those weights as an initialization to optimise for a target task using the target data. Our setting differs from this in that we do not utilise the data of the target task to find a solution. This is possible as the information of the target task is entirely contained in the source task. One could consider a setting where the source task only contains partial information of the target task, and thus does not provide the correct parameters directly, but only a good initialization which is then used to adapt to a target task.
>
> **Page 4: The language around functionals is confusing. You call f and g functionals, but I think they are standard functions? You also say that T is a map from function to function, but then you do function composition between T and f, which implies that T is a function from Y to Y (where Y is the codomain of f).**
>
> We thank the reviewer for pointing out these typos, we have updated the notation used in the paper.

---

### Official Review · Reviewer_VvbW · 2022-10-28

**Confidence:** 3
**Correctness:** 4
**Technical Novelty And Significance:** 3
**Empirical Novelty And Significance:** 3
**Recommendation:** 6

**Clarity, Quality, Novelty And Reproducibility:**

Clarity, quality, and reproducibility are good enough.

However, I'm not really sure if this the novelty of this paper is significant. There may be some other literatures solving the same problem, so I will defer it to the discussion phase.

**Strength And Weaknesses:**

Strength
- The paper is well written
- The problem formulation is novel and important
- The method is simple and intuitive
- The proposed method outperforms the simple baselines over various problems
- The proposed method provides a new way to generalize to a distributional shift

Weaknesses
- As already pointed out in the main paper, the dimensionality of both the parameters of the function f (source) and g (target) are high-dimensional, which may prevent the proposed method from being applied to larger scale problems. As far as I understand, the experiments are all small scale due to this reason.
- I wonder whether there exists no such literatures solving exactly the same problem. It's quite surprising. Maybe there should be some that I'm not aware of. I will defer this point to the discussion phase with other reviewers.
- It's unclear why the proposed method should outperform with strong noise level. Could you provide some intuition?
- Although it is nice that the method provides a way to generalize to a distributional shift, it is only for a single specific distributional shift at a time. For instance, in pendulum experiments it is only generalizable to double pendulum, while we would wish to generalize to any number of pendulums.

**Summary Of The Paper:**

This paper proposes to solve support-query distributional shift problem which has not been addressed by the previous meta-learning literatures. Instead of assuming that the same function f is used to sample both support and query set, they assume that different functions f and g are generating each support and query set. And then they propose to learn to map from f to g with simple transformations. The proposed method outperforms the simple baselines over various problems, especially well when the white noises are added.

**Summary Of The Review:**

In summary, the paper addresses a very important problem of support-query distributional shift that has not been fully addressed by the previous meta-learning literatures. The method is clear, simple, and effective. I thus recommend acceptance, but for the novelty part I will have a discussion with other reviewers.

---

> ### Author Response · Authors · 2022-11-16
> **Reply to reviewer VvbW**
>
> We thank the reviewer for the valuable comments and we would like to expand on some of the questions asked.
>
> **As already pointed out in the main paper, the dimensionality of both the parameters of the function f (source) and g (target) are high-dimensional, which may prevent the proposed method from being applied to larger scale problems. As far as I understand, the experiments are all small scale due to this reason.**
>
> It is true that when using the non-linear or linear method (eq 5 and 6), the full dimensionality of the parameter space must be taken into account. In practice, one could instead consider calculating the gradient over only a subset of the parameters (as in ANIL [1]) to reduce the dimensionality. This would also hold for the contextual method (eq 4) which finds the gradients w.r.t a low-dimensional conditioning variable. Extending the experiments to high-dimensional problems such as images is an avenue for future work.
>
> [1] Rapid Learning or Feature Reuse? Towards Understanding the Effectiveness of MAML
>
> **I wonder whether there exists no such literatures solving exactly the same problem. It's quite surprising. Maybe there should be some that I'm not aware of. I will defer this point to the discussion phase with other reviewers.**
>
> We have highlighted some of the most relevant works in the related work section. There are those that consider a shift in the domain and also examine problems from control such as Xian et al [2], Sanchez-Gonzalez [3]. These however do not account for a shift in the co-domain as well.
> There are also works that use gradients as task representations. For example [4] considers a similar setting to ours in that support and query come from different tasks but only considers problems from the NLP domain. In contrast to ours, they use a pre-trained base model $f_{\theta}$ to find the gradient representations, which are computed as the trace of the fisher-information matrix.
> Our paper considers problems in control and also show the benefits w.r.t noise in the input. We will add this to the related work.
>
> [2] Hyperdynamics: Meta-learning object and agent dynamics with hypernetworks
>
> [3] Graph networks as learnable physics engines for inference and control
>
> [4] Grad2Task: Improved Few-shot Text Classification Using Gradients for Task Representation
>
>
> **It's unclear why the proposed method should outperform with strong noise level. Could you provide some intuition?**
> This is a very good question. Past work such as [4] and [5] utilises gradient information to find task representations which reinforces the intuition of the paper. However, providing a thorough analysis is limited by the current understanding of deep neural networks and their stability w.r.t small perturbations in the input and parameters. This is a clear line of future work which we aim to explore further.
>
> [5] Task2Vec: Task Embedding for Meta-Learning
>
> **Although it is nice that the method provides a way to generalize to a distributional shift, it is only for a single specific distributional shift at a time. For instance, in pendulum experiments it is only generalizable to double pendulum, while we would wish to generalize to any number of pendulums.**
> We agree with the reviewer that this would be a very interesting property. This, however, would be possible by training our model on a general dataset of pendulums with the additional supervision of the number of pendulums or require a further form of adaptation to regress the number of pendulums at hand. We feel like this would be somewhat outside the scope of the work and decided to restrict the analysis to transfer between single and double pendulum.

---

### Decision · Program_Chairs · 2023-01-20

**Decision:**

Reject

**Justification For Why Not Higher Score:**

Given the scores and the fact that no reviewer has championed the paper, it is clear that the general opinion is that this paper is not ready for publication in ICLR. The key reasons are that (a) there is insufficient experimental evaluation (b) the scope of the method mostly in the experiments is unnecessarily constrained given what in theory such a method could achieve. Overall, the narrative of the paper is still a little confusing and not convincing enough.

**Justification For Why Not Lower Score:**

N/A

**Metareview: Summary, Strengths And Weaknesses:**

This paper introduces the idea of performing meta-learning when the support and query sets come from separate but related distributions, which can manifest as a transfer learning problem. Consequently, the authors assume different functions f and g for each set and create an encoder-decoder architecture for learning how to map from one domain to another.

Four expert reviewers read this paper and generally that the key idea is novel and intuitive. Apart from the utility of using it for cases where a distribution shift does manifest, it is also interesting to see this connection between meta-learning and transfer learning. The paper is also well organized and easy to read.

On the other hand, some obvious extensions of the paper -both in terms of experiments and theoretical discussion- are left for future work, see e.g. reviewer Vvbw's suggestion for generalization to multiple distribution shifts and ability to handle large scale problems with a subset of parameters or other dimensionality reduction method. These ideas feel like they should be in the core of the method rather be treated as extensions, since they constrain the scope of the method a bit too much.

On the same note, the reviewers generally agree that the experimental evaluation is rather limited. For one, the chaotic system induced by the double pendulum forces the authors to consider a small time-scale. Perhaps that's not the best example to use; for example, how this timescale has been selected, why not slightly smaller or larger? I would also argue that the pendulum example is still a synthetic experiment, given the artificial formulation of the task and the temporal constraint. Finally, it would be needed to include more baselines (as mentioned by reviewer 9QJw and 8eQb) and potentially even datasets: this is a relatively general application domain so to extract conclusions that support the claims of the paper it would be needed to have these additional experiments.

Finally, a majority of reviewers remain confused about what the transfer setting is, as it is a nonstandard definition (e.g. comments by Reveiwers VvbW and 9QJw). This affects the motivation and overall narrative of the paper. In a sense, the key idea is interesting and it does feel like it is general enough, but the exposition and experiments only expore a much more narrow view of what could actually be considered. That directly links to the comment above for obvious extensions left for future work and limited experiments.